# ALARA in Pediatric Electrophysiology Laboratory

**DOI:** 10.3390/children9060866

**Published:** 2022-06-10

**Authors:** Harinder R. Singh

**Affiliations:** Baylor College of Medicine, Children’s Hospital of San Antonio, San Antonio, TX 78207, USA; harinder.singh@bcm.edu; Tel.: +210-7044789

**Keywords:** electrophysiology, “as low as reasonably achievable” (ALARA), fluoro-less, electroanatomical mapping

## Abstract

The effects of radiation on patients and the providers are dose-dependent and cumulative. Pediatric patients are not only more sensitive to radiation but also may undergo more procedures and diagnostic tests throughout their lifetime. As providers, the endeavor is to cause no harm and it behooves us to either eliminate or minimize the radiation exposure to patients without affecting the efficacy and outcomes of the diagnostic or therapeutic modalities. Pediatric electrophysiologists have taken the lead in attempting to minimize the radiation exposure to patients and staff with innovative and advanced techniques. The techniques range from minimizing the exposure to radiation with better understanding and applications of the physics associated with fluoroscopic imaging to using alternative imaging modalities that do not use radiation.

## 1. Introduction

Advances in electrophysiology (EP) have been monumental with great strides in better understanding of the mechanism and interpretation of the data as well as management options. Changes in adult EP related to understanding the physiology and management strategies precede the changes in the pediatric EP practice, most likely dictated by the larger volume of patients in the adult practice. One of the avenues where pediatric EP has been at the cutting edge is in developing strategies to either minimize or eliminate fluoroscopic exposure. This has been triggered by the concerns with radiation exposure to patients as well as the staff. Pediatric patients seem to be more sensitive to radiation than adults and may undergo more procedures and imaging during their lifetime, making the risk of exposure cumulative. 

## 2. Background

There have been recognized dose-dependent and cumulative effects of radiation on the patients, the operators, as well as laboratory personnel. These include cataracts, skin conditions, malignancies, and birth defects, as well as effects related to the protective equipment in the form of spine and orthopedic injuries, leading to long-established standards for occupational exposure policy of “as low as reasonably achievable (ALARA)”. The goal of the policy is to eliminate all unnecessary radiation exposure to patients, physicians, and personnel, as well as provide adequate imaging for diagnosis and intervention. The Multi-Specialty Occupational Health Group (MSOHG) was established to define the occupational risks associated with working in the fluoroscopic laboratories. Member organizations of the MSOHG include the Society of Cardiac Angiography and Interventions, Society of Interventional Radiology, Heart Rhythm Society, American College of Radiology, American College of Cardiology, Society of Neuro-Interventional Surgery, American Association of Physicists in Medicine, and Society of Invasive Cardiac Professionals [1]. Known factors that influence exposure to the radiation dose during procedures include patient complexity, physician body size, height, gender, experience, and skill level, as well as workload, procedure technique, position in relationship with the patient, awareness, and protective equipment. Some of these factors are modifiable to achieve the concept of ALARA [2]. Beyond doubt, the most effective method would be the elimination of radiation if feasible. To this extent, the use of multiple techniques and advanced imaging has revolutionized the approach to achieve ALARA radiation exposure without compromising the safety and efficacy of interventional and electrophysiological procedures.

## 3. Historical Landmarks

Cardiac mapping was first studied by Lewis et al. [3] in dogs in 1915. Gepstein et al. [4] described a method for a non-fluoroscopic catheter based endocardial electroanatomical mapping of the swine heart in 1997, using a mapping and navigation system comprised of a miniature passive magnetic field sensor, an external ultralow magnetic field emitter, and a processing unit (CARTO, Biosense, Irvine, CA, USA). Smeets et al. [5] reported the first use of this method for non-fluoroscopic endocardial mapping in 15 patients in 1998. Drago et al. [6] described their experience in 21 children with WPW syndrome due to right-sided accessory pathways, which was the first reported pediatric experience using the non-fluoroscopic technique in 2002. Smith et al. [7] described their experience with 30 pediatric patients with total elimination of fluoroscopy and using a three-dimensional mapping system in 2006. Bharmanee et al. [8] reported the feasibility, safety, and accuracy of 3D electroanatomic mapping without the use of fluoroscopy in patients with congenital heart defects (CHDs).

## 4. Strategies

Strategies to reduce exposure to radiation include measures and techniques to decrease radiation when fluoroscopic procedures are performed as well as supplementation with techniques that can either reduce or replace the need for fluoroscopy. Advances in mapping techniques and advanced imaging have facilitated significant reduction in fluoroscopy use in pediatric electrophysiology laboratories. These strategies and modalities include:

1. Strategies to reduce the radiation exposure when using fluoroscopy: The ideal strategy would be to eliminate the use of radiation; however, this is not always feasible. The various suggestions include judicious use of radiation only when imaging is necessary, minimizing the use of cine, avoiding steep angles of the X-ray beam, minimizing the use of magnification modes, minimizing the frame rate of fluoroscopy and cine, keeping the image receptor close to the patient, utilizing collimation to the fullest extent possible, monitoring of radiation dose in real time, using and maintaining appropriate protective garments, maximizing distance of operator from the X-ray source and patient, using protective shields in the optimal position at all times, keeping the table height as high as comfortably possible for the operator, varying the imaging beam angle to minimize exposure to any one skin area, and keeping the patient’s extremities out of the beam [9]. We tend to use all the above features as feasible, but most importantly we do not use cine unless necessary, decrease the frame rate to start with three frames per second and gradually increase if needed, maintain appropriate distance between the patient and the detector, use collimation as well as use an anti-scatter grid if the patient is greater than 25 kg, and only expose the site of interest to the fluoroscopy beam. All the personnel in the laboratory use appropriate garments and maintain a fair distance from the source of radiation.

2. Three-dimensional mapping: Three-dimensional electroanatomic mapping systems can be used as an adjunct to conventional mapping to help reduce the exposure to radiation, and they can also be used primarily to help with mapping of arrhythmias where they can display the real-time position of catheters on computerized constructed three-dimensional cardiac chamber anatomy. They can provide activation timing and voltage mapping as well as annotate and tag lesions. Of the various platforms that are available, currently the predominantly used systems include the CARTO-based navigation system (Biosense Webster Inc., Irvine, CA, USA), Ensite NavX platform (Saint Jude Medical, St. Paul, MN, USA; now Abbott Laboratories, Chicago, IL, USA), and Rhythmia (Boston Scientific, Cambridge, MA, US). 

The CARTO system uses magnetic fields for electroanatomical navigation and reconstruction of geometry. For mapping, this system traditionally has required dedicated manufactured catheters that have magnetic sensors at their tip. It can visualize nondedicated catheters using impedance-based navigation but without the possibility to use these catheters for the reconstruction of geometry or tagging local lesions on the maps. The current version the CARTO Prime mapping module incorporates Coherent mapping for the diagnosis of scar-related complex atrial arrhythmias, Cartofinder for identifying repetitive focal and rotational activities, Parallel mapping to allow for simultaneous mapping of different arrhythmias using the same catheter locations, and LAT Hybrid to improve location accuracy [10]. This system can merge computed tomography or magnetic resonance imaging reconstructed segmental cardiac chambers of interest with the created map. It also provides the ability to use intracardiac echocardiographic-based 3D reconstruction and integration of rotational angiography or fluoroscopy views as well as automated annotation of locations of radiofrequency applications. 

The Abbott EnSite Precision (NavX mode) (Abbott Laboratories, Chicago, IL, US) is an impedance-based navigation system that generates an electric current from patches placed around the patient’s torso and collects impedance measurements from electrodes on catheters introduced into the body. NavX is considered an open-platform system that can visualize any catheter with electrodes and collect mapping data as soon as the catheters are introduced into the vasculature, facilitating seamless fluoro-less procedure workflows. However, impedance-based systems can be sensitive to environmental and anatomical impedance fluctuations. EnSite Precision and newer EnSite platforms are hybrid impedance-magnetic systems, which also collect magnetic data from sensor-enabled catheters to optimize the geometric linearity of the model. Furthermore, segmented cardiac CTs and MRIs can be imported into a study, visualized next to the collected geometry, and fused with the electroanatomical map. EnSite X, Abbott’s newest mapping system release, offers both a primary impedance mode (NavX) and a primary magnetic mode (VoXel). Recent additions in the form of contact force mapping, use of high-density grid catheters, and the ability to merge the 3D reconstructed image from a CT scan or MRI helps in achieving accurate 3D geometry, improving mapping, and obtaining activation mapping and voltage mapping to define scar tissue and tag lesions [11]. Non-contact mapping using an Ensite Array catheter allowed recording of far field unipolar potentials distant from the endocardial surface, reconstructing virtual electrograms on the endocardial surface, useful for non-sustained and non-tolerated arrhythmias, but currently it may not be available.

The Rhythmia system (Boston Scientific, (Cambridge, MA, US) also uses both magnetic and impedance-based localizing techniques to create a 3D map with very high resolution. However, it requires initially creating a map using dedicated magnetic catheters, and subsequently the standard catheters can be visualized. It reduces the amount of interpolation between annotated points to more efficiently identify areas of interest, clearly visualize propagation of complex arrhythmia circuits, and characterize complex substrates, including critical isthmuses, low-voltage regions of interest, and scar and scar boundaries with improved accuracy using high density mapping catheters with flat electrodes [12].

3. Magnetic navigation system/robotic technology, (MNS, Niobe, Stereotaxis, St. Louis, MO, USA): Stereotaxis uses magnets to control the catheter tip and imaging to guide movements, allowing for more precise and safer navigation than traditional mechanical catheters. The catheters can be navigated remotely with the magnetic guide, taking away the risk of injury as well as achieving more precise and consistent contact, allowing for optimal mapping and ablation. It has been used for ablation of ventricular arrhythmias and atrial fibrillation and has been reportedly used in congenital heart defects as well [13]. 

4. Advanced imaging as an adjunct: Real-time 3-dimensional trans-esophageal echocardiography (3D TEE) provides enough spatial and temporal resolution to track the movements of catheters and greatly improve navigation of the catheters. Intracardiac echocardiography (ICE) has been used to perform trans-septal punctures as well as placement of catheters.

## 5. Techniques

The measures to reduce the exposure to radiation have been described above. Here, we will discuss the technique used for electroanatomical mapping to reduce or eliminate the use of fluoroscopy. Based on what mapping system is being used for the procedure, the patches are placed on the patient accordingly. The mapping catheters are advanced from the venous access site, usually the femoral vein. As soon as atrial electrograms are recorded, the (Inferior vena cava- right atrium) IVC-RA junction is marked. A point cloud is collected as the catheter is maneuvered in the cardiac chamber. The catheter is advanced, and when the atrial electrograms are no longer seen, the (Superior vena cava-right atrium) SVC-RA junction is marked. The catheter is withdrawn and, utilizing the intracardiac electrograms and the orientation of the catheter, the HIS potential is marked, and the catheter is shadowed for future reference. The catheter is manipulated to delineate the TV annulus based on equal amplitudes of atrial and ventricular electrograms. The catheter is then maneuvered into the coronary sinus. Once the geometrical details are obtained, other catheters are placed and moved around in the chamber, with real-time visualization in any angulated projection and magnification (Figure 1 and Figure 2). The electrophysiology study is then performed per protocol. The details for ventricular geometry are usually limited to the site of interest (Figure 3).

A trans-septal puncture can also be performed using the mapping system as well as with guidance using TEE or ICE [14,15]. Some EP physicians still prefer using fluoroscopic assistance, as the exposure to radiation is limited. Similarly, once the catheter is placed in the left atrium, geometry is typically collected continuously so that the site of the trans-septal puncture can be marked and identified for re-entering the left atrium again, if required.

## 6. Outcomes

The understanding of risks of radiation, improvement, and advances in technology as well as skill levels has led to immense progress in reducing the exposure to radiation. From two decades earlier, the electroanatomic mapping used as an adjunct to fluoroscopy has steadily evolved to the current era where it is used primarily for EP studies and ablation with the adjunct of either echocardiographic or brief fluoroscopy exposure during a trans-septal puncture to eliminate or significantly reduce the exposure to radiation. 

Over the last 15 years, the use of the non-fluoroscopic electroanatomic mapping systems has expanded, along with access to the technology in most contemporary EP laboratories. These systems have led to a significant decrease in radiation exposure, and procedures can be performed with minimal or zero radiation exposure [6,7,16,17,18,19,20,21]. Transesophageal or intracardiac echocardiography (ICE) can also be used along with these 3D mapping systems to help visualize and define anatomy and catheters, with or without integration of the imaging modalities [16,20]. The efficacy and safety of non-fluoroscopic imaging for straightforward SVT ablation in pediatrics has been evaluated in retrospective studies [6,7,16,18,19,22] and prospective randomized controlled studies [17,20]. When comparing adjunct use of non-fluoroscopic imaging systems with fluoroscopy alone, the reported success rates, long-term recurrence rates, and complication rates are identical to the use of fluoroscopy alone, with significantly reduced exposure to radiation, irrespective of the type of mapping system used. Based on the available data, non-fluoroscopic imaging systems reduce radiation exposure and can be safely and effectively used for most SVT ablations in pediatric patients without putting the patients at any additional risks. Ultrasound is generally the only non-fluoroscopic imaging technology demonstrated to safely reduce the need for fluoroscopy during trans-septal puncture.

Catheter ablation for SVT in children can be successfully performed in patients with normal cardiac anatomy using minimal or no fluoroscopy with favorable outcomes [23]. Similarly, the success and complication rates were uncompromised using the ALARA protocol for reducing the radiation exposure [24]. This technique of eliminating the need for fluoroscopy has also led investigators to consider performing EP studies and ablation in patients with uncomplicated arrhythmias, in a non-traditional setup with similar outcomes and results without encountering procedural difficulties [25]. This may open the options of performing these procedures in the intensive care rooms or procedure suites. The fluoro-less or near-zero fluoroscopy approach has been used in patients with normal cardiac structure and anatomy predominantly. However, it has been reliably shown to be not only feasible but also safe and accurate in children and young adults with various congenital heart defects (Figure 4) as long as there are no intracardiac leads, prosthetic valves, or extensive scarring [8].

## 7. Adverse Effects 

The safety of the fluoro-less or near-zero fluoroscopic procedures has been proven; however, there have been a few reports of some complications that could have been prevented if fluoroscopy was used. A case report on the entrapment of a diagnostic mapping catheter within an Advisor HD grid catheter [26] as well as severe pulmonary regurgitation secondary to ablation in the proximal main pulmonary artery (MPA) to ablate the focus of PVCs that appeared to be from right ventricular outflow tract (RVOT) [27] highlight the need for the availability of fluoroscopy during the performance of fluoro-less or near-zero fluoroscopic procedures. 

## 8. Conclusions

Based on the 2016 HRS statement, it is a class I safety recommendation to use fluoroscopy as low as reasonably achievable (ALARA) for all ablation procedures in pediatric and congenital cases [28]. Fluoro-less or near-zero fluoroscopy procedures have been demonstrated with uncompromised acute and long-term success and complication rates when compared with the procedures performed with fluoroscopy guidance. There are also data about the safe and efficacious use of fluoro-less procedures in certain more complex patients with significant congenital heart defects. These can be performed with optimal use of 3D electroanatomical mapping primarily or as an adjunct and with the aid of advanced imaging, if required. The advantages of fluoro-less or near-zero fluoroscopic procedures include the avoidance of unnecessary exposure of patients and healthcare providers, improving the longevity of careers, the ability for the pregnant staff members to continue working in the EP laboratory, and the ability to perform procedures on special groups such as pregnant women.

## Figures and Tables

**Figure 1 children-09-00866-f001:**
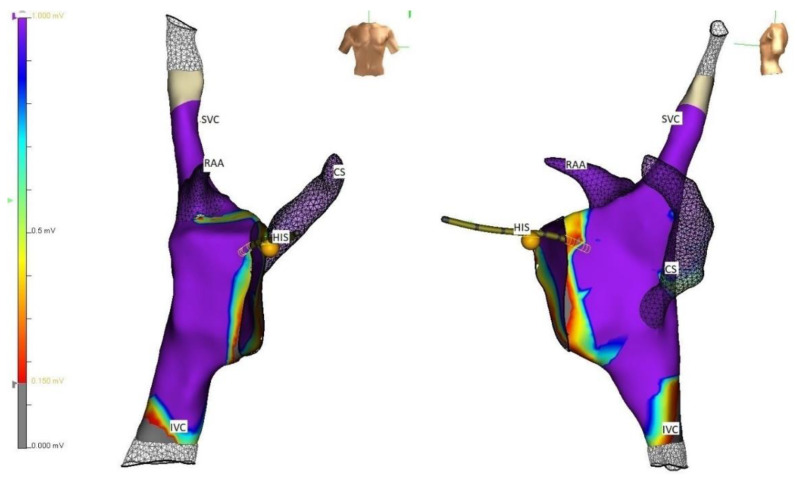
Three-dimensional electroanatomical mapping showing the right atrial geometry with voltage and labelling of structures in straight frontal and lateral projection. RAA: right atrial appendage; CS: coronary sinus, HIS: His bundle; SVC superior vena cava; IVC: inferior vena cava.

**Figure 2 children-09-00866-f002:**
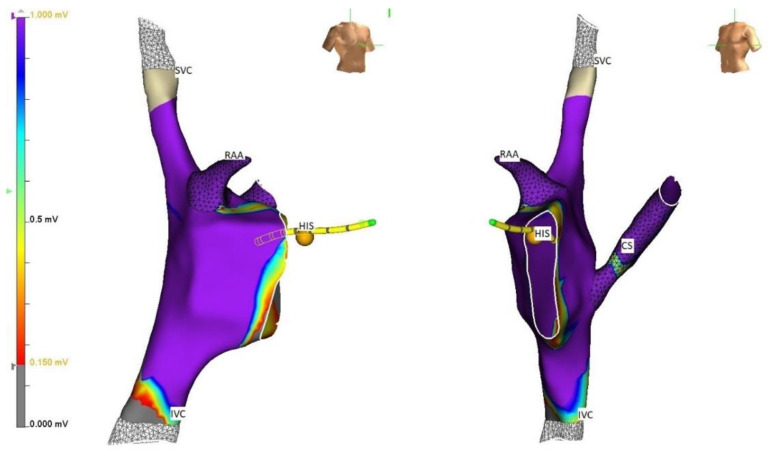
The same geometry and anatomy as seen in Figure 1, in right anterior oblique (RAO) and left anterior oblique (LAO) projection highlighting the ability to see the catheters and geometry in the desired projection.

**Figure 3 children-09-00866-f003:**
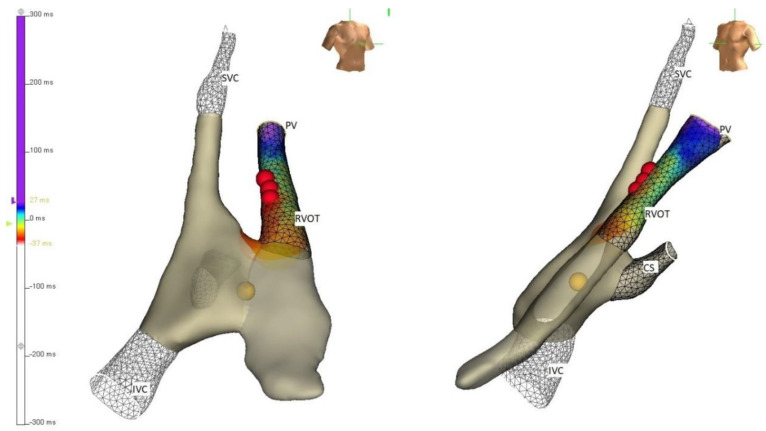
Three-dimensional electroanatomical mapping of the right ventricular outflow tract in RAO and LAO projection limited to the site of interest for mapping and ablation of the premature ventricular complexes (PVCs). RVOT: right ventricular outflow tract.

**Figure 4 children-09-00866-f004:**
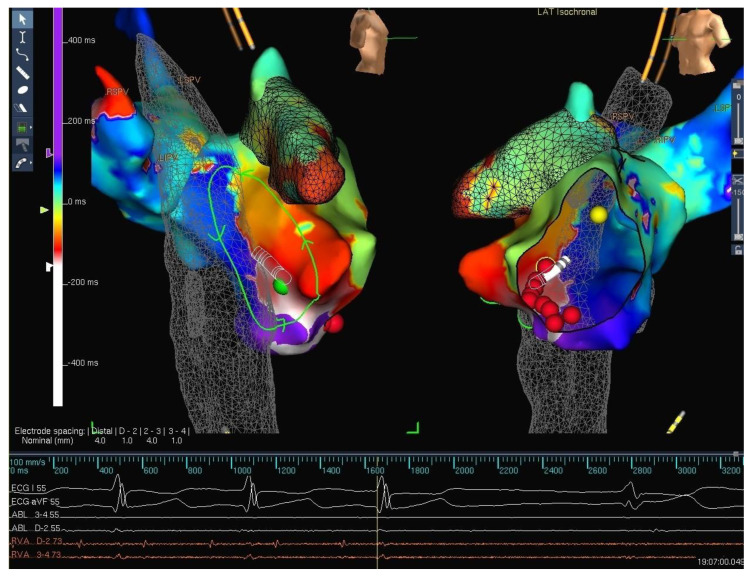
Three-dimensional electroanatomical mapping of a patient with single ventricular physiology post Fontan palliation with intra atrial re-entrant tachycardia evident from the activation mapping. The non-colored part of the mapping shell is representative of the lateral tunnel. The non-standard projection in this image is to show the important part of the circuit and geometry, again indicating the ability to view the geometry from any projection as deemed necessary.

## Data Availability

Not applicable.

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
