# Peer review of "ALARA in Pediatric Electrophysiology Laboratory"

_children, 2022, doi:10.3390/children9060866_

Round 1

Reviewer 1 Report

I've checked the revised manuscript. The authors have addressed my concern, the manuscript has been improved. In my opinion, it is suitable to publish in "Children".

Author Response

Thanks so much.

Reviewer 2 Report

The manuscript is a brief review on the low fluoro/fluoroless approach to ablation in the pediatric population. I have no further recommendations.

Author Response

Thanks. Again the aim of the paper was a review and therefore did not pursue the meta analysis.

Reviewer 3 Report

The author reviewed a low or no fluroscopic approach to performing catheter ablation in the paediatric population. The various methods are well described. It will be most useful to provide a meta-analysis, reporting on the success rates, procedure time, ionising doses and fluroscopic times.

Author Response

As this was a review article, we did not venture into performing a meta analysis however most of the state of the art papers related to the topic are referenced and would help the readers to further analyze each article as deemed fit.

This manuscript is a resubmission of an earlier submission. The following is a list of the peer review reports and author responses from that submission.

Round 1

Reviewer 1 Report

The authors described the technique for the policy "as low as reasonably achievable" of the radiation exposure. The main technique is used the electrophysiological method for mapping the 3-D of the human organ, such as hart. It is very interesting, my comment is described below.

How much could the radiation exposure be reduced by using this technique? The description of the reference and/or theoretical values would help the readers better understanding.

Reviewer 2 Report

The manuscript briefly reviews the current state of the art of fluoroless cardiac ablation in the pediatric population. After a brief historical review of cardiac mapping, currently available mapping systems are described. The technical aspects and process of fluoroless EP study/ablation is only very briefly described. The discussion ("outcomes") lists some studies which analyzed the safety and efficacy of low fluoro/fluoroless ablation, however, no attempts were made to process, analyze and present the data in detail. Without a more detailed analysis, this brief review does not add significant new information to previously published articles.

Reviewer 3 Report

The authors provided a succinct description of ALARA approaches in the Pediatric EP Laboratory.

My comments

1. Please expand the ways the operator may reduce ionising radiation doses should fluoroscopy be utilised, eg reduce frame rates, doses, etc.